# LP-3DGS: Learning to Prune 3D Gaussian Splatting

**Zhaoliang Zhang**
Johns Hopkins University
Baltimore, MD 21218
zzhan288@jh.edu

**Tianchen Song**
Johns Hopkins University
Baltimore, MD 21218
tsong15@jh.edu

**Yongjae Lee**
Arizona State University
Tempe, AZ 85281
ylee298@asu.edu

**Li Yang**
University of North Carolina at Charlotte
Charlotte, NC 28223
lyang50@uncc.edu

**Cheng Peng**
Johns Hopkins University
Baltimore, MD 21218
cpeng26@jhu.edu

**Rama Chellappa**
Johns Hopkins University
Baltimore, MD 21218
rchella4@jhu.edu

**Deliang Fan**
Arizona State University
Tempe, AZ 85281
dfan@asu.edu

## Abstract

Recently, 3D Gaussian Splatting (3DGS) has become one of the mainstream methodologies for novel view synthesis (NVS) due to its high quality and fast rendering speed. However, as a point-based scene representation, 3DGS potentially generates a large number of Gaussians to fit the scene, leading to high memory usage. Improvements that have been proposed require either an empirical pre-set pruning ratio or importance score threshold to prune the point cloud. Such hyperparameters require multiple rounds of training to optimize and achieve the maximum pruning ratio while maintaining the rendering quality for each scene. In this work, we propose learning-to-prune 3DGS (LP-3DGS), where a trainable binary mask is applied to the importance score to automatically find a favorable pruning ratio. Instead of using the traditional straight-through estimator (STE) method to approximate the binary mask gradient, we redesign the masking function to leverage the Gumbel-Sigmoid method, making it differentiable and compatible with the existing training process of 3DGS. Extensive experiments have shown that LP-3DGS consistently achieves a good balance between efficiency and high quality.

## 1 Introduction

Novel view synthesis (NVS) takes images and their corresponding camera poses as input and seeks to render new images from different camera poses after 3D scene reconstruction. Neural Radiance Fields (NeRF) (Mildenhall et al. [2021]) uses multi-layer perceptron (MLP) to implicitly represent the scene, fetching the transparency and color of a point from the MLPs. NeRF has gained considerable attention in the NVS community due to its simple implementation and excellent performance. However, in order to obtain a point in the space, NeRF needs to perform an MLP inference. Each pixel rendered requires processing a ray and there are many sample points along each ray. Consequently, rendering an image requires a large amount of MLP inference operations. Thus, rendering speed becomes a major drawback of the NeRF method.

38th Conference on Neural Information Processing Systems (NeurIPS 2024).

Besides NeRF, explicit 3D representations are also widely used. Compared to NeRF, the advantage of point-based scene representation is that it is well-supported by modern GPU rendering, enabling fast render speed. 3D Gaussian Splatting (3DGS) (Kerbl et al. [2023]) achieves both good quality and high rendering speed, making it a hot topic in the community. 3DGS uses 3D Gaussian models with color parameters to fit the scene and develops a training framework to optimize the model parameters. However, the number of points required to reconstruct the scene is enormous, usually in the millions. In practice, each point has dozens of floating-point parameters, which makes 3DGS a memory-intensive method.

Some recent works have attempted to mitigate this problem by pruning the points, such as Light-Gaussian (Fan et al. [2023]), RadSplat (Niemeyer et al. [2024]), and Mini-Splatting (Fang and Wang [2024]). These methods follow a similar pruning approach through defining an *importance score* for each Gaussian point and prune the points with such importance score below a preset empirical threshold. However, a major drawback of these methods is that the preset threshold must be manually tuned through multiple rounds of training process to identify the favorable pruning ratio to minimize the number of Gaussian points while keeping the rendering quality. To make matters worse, such number of points may vary depending on different scenes, which requires manual pruning ratio searching for each scene. For example, the blue and red lines in Figure 1 show the rendering quality of *Kitchen* and *Room* scenes, respectively, in MipNeRF360 scenes (Barron et al. [2022]), with sweeping 12 different pruning ratios (i.e., 12 rounds of training) following the prior RadSplat (Niemeyer et al. [2024]) and Mini-Splatting (Fang and Wang [2024]) method. It is clearly seen that a smaller pruning ratio will not hamper the rendering quality, and the rendering quality will start to decrease with much more aggressive pruning ratios. Both scenes exhibit an optimal pruning ratio region that could maximize the pruning ratio and maintain the rendering quality. It could also be seen that the ideal pruning ratio differs for these two scenes.

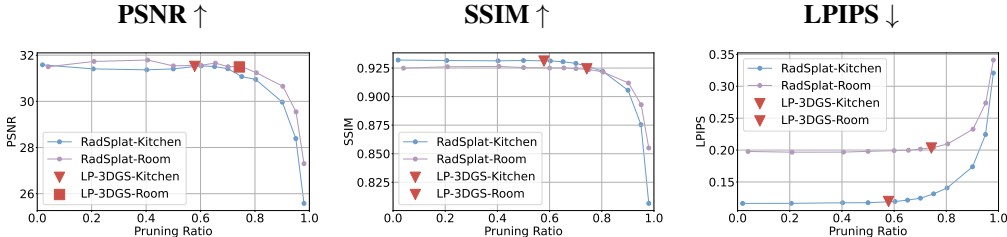

Figure 1: The performance changes with the pruning ratio of RadSplat on the MipNeRF360 dataset *Kitchen* and *Room* scenes are shown in blue and purple lines, respectively. Red triangles and squares represent the results of LP-3DGS on the importance score of RadSplat. **LP-3DGS is able to find the favorable pruning ratio in one training session instead of requiring dozens of attempts to find the best hyperparameter**.

In this paper, we propose a learning-to-prune 3DGS (LP-3DGS) methodology where a trainable mask is applied to the importance score. Notably, it is compatible with different types of importance scores defined in prior works. Instead of a preset threshold to determine the 3DGS model pruning ratio, as illustrated by the red triangle symbol in Figure 1, our method aims to integrate with existing 3DGS model training process to learn the favorable pruning ratio for minimizing the model size while maintaining the rendering quality. Since the traditional hard-threshold-based binary masking function is not differentiable, a recent prior work, Compact3D (Lee et al. [2023]), leverages the popular straight through estimator (STE) (Bengio et al. [2013]) to bypass the mask gradient for adaption to the backpropagation process. However, this approach often results in suboptimal pruning ratios. In contrast, in our work, we propose to redesign the masking function leveraging the Gumbel-Sigmoid activation function to make the whole masking function differentiable and integrate with the existing training process of 3DGS. Consequently, LP-3DGS automatically minimizes the number of Gaussian points for each scene with just a single round of training.

In summary, the technical contributions of our work are:

- To address the effortful tuning of 3DGS pruning ratios, we propose a learning-to-prune 3DGS (LP-3DGS) methodology that leverages the differentiable Gumbel-Sigmoid activation function to embed a trainable mask with different types of existing importance scores

designed for pruning redundant Gaussian points. As a result, instead of fixed model size, LP-3DGS could learn an favorable Gaussian point size for individual scene with only one-time training.

- We conducted comprehensive experiments on state-of-the-art (SoTA) 3D scene datasets, including MipNeRF360 (Barron et al. [2022]), NeRF-Synthetic (Mildenhall et al. [2021]), and Tanks & Temples (Knapitsch et al. [2017]). We compared our method with SoTA pruning methods such as RadSplat (Niemeyer et al. [2024]), Mini-Splatting (Fang and Wang [2024]), and Compact3D (Lee et al. [2023]). The experimental results demonstrate that our method can enable the model to learn the favorable pruning ratio and that our trainable mask method performs better than the STE mask.

## 2 Related Work

**Neural radiance fields (NeRFs)**    NeRFs (Mildenhall et al. [2021]) targets to represent the scene in multilayer perceptrons (MLPs) based on multi-view image inputs, enabling high-quality novel view synthesis. Due to its advancement, numerous follow-up works improved it in either rendering quality (Barron et al. [2021, 2022]) or efficiency(Müller et al. [2022], Chen et al. [2022], Fridovich-Keil et al. [2022]).

Although NeRF models demonstrate impressive rendering capabilities across numerous benchmarks, and considerable efforts have been made to enhance training and inference efficiency, they typically still face challenges in achieving fast training and real-time rendering.

**Radiance Field Based On Points.**    In addition to implicit representations, several works have focused on volumetric point-based methods for 3D presentation (Gross and Pfister [2011]). Inspired by neural network concepts, (Aliev et al. [2020]) introduced a neural point-based approach to streamline the construction process. Point-NeRF (Ding et al. [2024]) further applied points for volumetric representation, enhancing the effectiveness of point-based methods in radiance field modeling.

**Gaussian Splatting**    3D Gaussian Splatting (3DGS) (Kerbl et al. [2023]) represents a significant advancement in novel view synthesis, utilizing 3D Gaussians as primitives to explicitly represent scenes. This approach achieves state-of-the-art rendering quality and speed while maintaining relatively short training time. A series of methods have been introduced to improve the rendering quality through using regularization for better optimization, including depth map (Chung et al. [2023], Li et al. [2024a]), surface alignment (Guédon and Lepetit [2023], Li et al. [2024b]) and rendered image frequency (Zhang et al. [2024]). However, the extensive number of Gaussians required for scene representation often results in a model that is too large for efficient storage. Recent research has focused on compression methods to enhance the efficiency of this representation. Notably, several studies (Fan et al. [2023], Fang and Wang [2024], Niemeyer et al. [2024]) have proposed using predefined scores as pruning criteria to keep Gaussians that significantly contribute to rendering quality. Compact3D (Lee et al. [2023]) introduces a method that applies a trainable mask on scale and opacity to each Gaussian and utilizes a straight-through estimator (Bengio et al. [2013]) for gradient updates. LightGaussian (Fan et al. [2023]) employs knowledge distillation to reduce the dimension of spherical harmonics. Additionally, (Fan et al. [2023], Lee et al. [2023]) also explored quantization techniques to further compress model storage. The previously proposed pruning methods primarily rely on predefined scores to determine the importance of each Gaussian. These approaches present two main challenges: first, whether the criteria accurately reflect the importance of the Gaussians, and second, the need for a manually selected pruning threshold to decide the level of pruning. In this work, we address these issues by introducing a trainable mask activated by a Gumbel-sigmoid function, applied to the scores derived from prior methods or directly to the scale and opacity of each Gaussian for more flexibility. Our approach automatically identifies the balance between the pruning ratio and rendering quality, eliminating the need to test on various pruning ratios.

## 3 Methodology

The conventional pruning methods leveraging predefined importance score require pruning ratio as a manually tuned parameter to reduce the size of Gaussian points in 3DGS. To seek for a favorable

pruning ratio, these methods may need to perform multiple rounds of training for each individual scene, which is inefficient. Motivated by this, we propose a **learning-to-prune 3DGS (LP-3DGS)** algorithm which learns a binary mask to determine the favorable pruning ratio for each scene automatically. Importantly, the proposed LP-3DGS is compatible with various types of pruning importance scores. In this section, we will: 1) introduce the preliminary of the original 3DGS and recap different importance metrics for pruning that are proposed by prior works, and 2) present the proposed learning-to-prune 3DGS algorithm.

### 3.1 3DGS Background

**3DGS Parameters**  3DGS is an explicit point-based 3D representation that uses Gaussian points to model the scene. Each point has the following attributes: position $\mathbf{p} \in \mathbb{R}^3$, opacity $\sigma \in [0, 1]$, scale in 3D $\mathbf{s} \in \mathbb{R}^3$, rotation represented by 4D quaternions $\mathbf{q} \in \mathbb{R}^4$ and fourth-degree spherical harmonics (SH) coefficients $\mathbf{k} \in \mathbb{R}^{48}$. In summary, one gaussian point has 59 parameters. The center point $X$ of a Gaussian model is represented by $\mathbf{p}$ and covariance matrix $\Sigma$ is denoted by $\mathbf{s}$ and $\mathbf{q}$. The SH coefficients model the color as viewed from different directions. The parameters of the Gaussians are optimized through gradient backpropagation of the loss between the rendered images and the ground truth.

**Rendering on 3DGS**  In order to render an image, the first step is projecting the Gaussians to 2D camera plane by world to camera transform matrix $W$ and Jacobian $J$ of affine approximation of the projective transform. The covariance matrix of projected 2D Gaussian is

$$\Sigma^{'} = JW\Sigma W^T J^T \tag{1}$$

The projected Gaussians would be rendered as splat (Botsch et al. [2005]), the color of one pixel could be rendered as

$$\mathbf{c_i} = \sum_{\mathbf{j=1}}^{\mathbf{N}} \cdot \mathbf{c_j} \cdot \alpha_{\mathbf{j}} \cdot \mathbf{T_j} \cdot \mathbf{G_j^{2D}} \tag{2}$$

Where $i$ is the pixel index, $j$ is the Gaussian index and $N$ is the number of the Gaussians in the ray. $c_j$ is the color of the Gaussian calculated by SH coefficients, $\alpha_j = (1 - exp^{-\sigma_j \delta_j})$, $\sigma_j$ is the opacity of the point and $\delta_j$ is the interval between points. $T_j = \prod_{k=1}^{j-1}(1 - \alpha_k)$ is the transmittance from the start of rendering to this point. $\mathbf{G_j^{2D}}$ is the 2D Gaussian distribution.

**Adaptive Density Control of 3DGS**  At the start of training, the Gaussians are initialized using Structure-from-Motion (SfM) sparse points. To make the Gaussians fit the scene better, 3DGS applies an adaptive density control strategy to adjust the number of Gaussians. Periodically, 3DGS will grow Gaussians in areas that are not well reconstructed, a process called "densification." Simultaneously, Gaussians with low opacity will be pruned.

### 3.2 Importance Metrics for Pruning

A straightforward way to prune the Gaussians is by sorting them based on a defined importance score and then removing the less important ones. Consequently, one of the main objective of prior 3DGS pruning works is to define an effective importance metric.

RadSplat (Niemeyer et al. [2024]) defines the importance score as the maximum contribution along all rays of the training images, written as

$$\mathbf{S_i} = \max_{\mathbf{I_f} \in \mathcal{I}_\mathbf{f}, \mathbf{r} \in \mathbf{I_f}} \alpha_\mathbf{i}^\mathbf{r} \cdot \mathbf{T_i^r} \tag{3}$$

Where $\alpha_i^r \cdot T_i^r$ is the contribution of Gaussian $G_i$ along ray $r$. RadSplat performs pruning by applying a binary mask according to the importance score, where the mask value for Gaussian $G_i$ is

$$m_i = m(\mathbf{S_i}) = \mathbb{1}(\mathbf{S_i} < \mathbf{t_{prune}}) \tag{4}$$

Where $t_{prune} \in [0, 1]$ is the threshold of score magnitude for pruning, $\mathbb{1}[\cdot]$ is the indicator function.

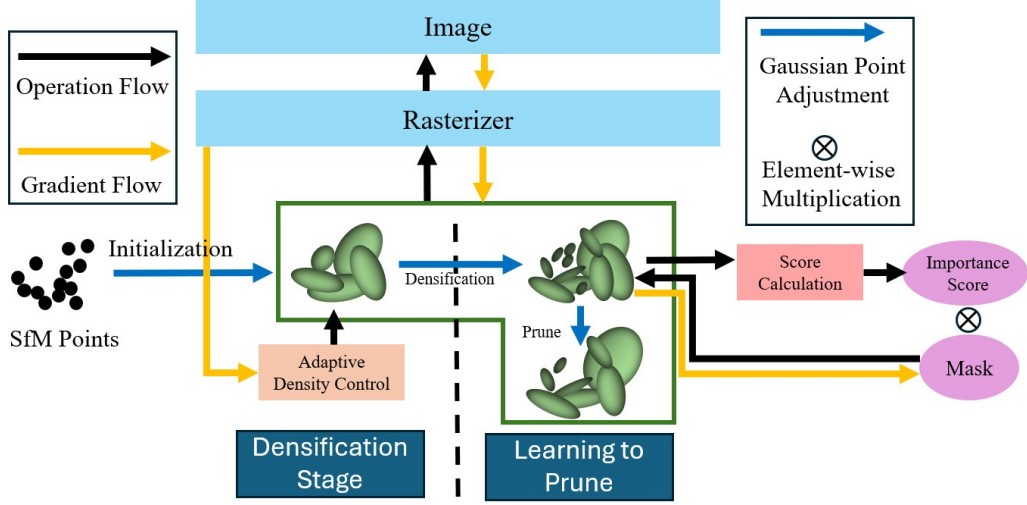

Figure 2: Overall learning process of the proposed LP-3DGS.

Another recent work, Mini-Splatting (Fang and Wang [2024]), uses the cumulative weight of the Gaussian as the importance score, which can be formulated as:

$$\mathbf{S_i} = \sum_{\mathbf{j=1}}^{\mathbf{K}} \omega_{\mathbf{ij}} \tag{5}$$

Where K is the total number of rays intersected with Gaussian $G_i$, $\omega_{ij}$ is the color weight of Gaussian $G_i$ on the $j$-th ray.

### 3.3 Learning-to-Prune 3DGS

The overall LP-3DGS learning process is shown in the Figure 2. In general, it mainly can generally be divided into two stages:1) densification stage, and 2) learning-to-prune stage. Following the original 3DGS, densification stage applies an adaptive density control strategy to gradually increase the number of Gaussians. As revealed by prior pruning works (Lee et al. [2023], Niemeyer et al. [2024]), 3DGS contains a significant number of redundant Gaussians. Subsequently, in the learning-to-prune stage, the proposed LP-3DGS learns a trainable mask upon a previously defined importance metric to compress the number of Gaussians with an favorable pruning ratio automatically. Specifically, to learn a binary mask, we first initialize a real-value mask $m_i$ for each point $i$, and then adopt the Gumbel-sigmoid technique to binarize the mask value differently.

**Gumbel-Sigmoid based Trainable Mask**   The binarization operation for real-value mask in pruning typically involves a hard threshold function, determining the binary mask should be 0 or 1. However, such hard threshold function is not differentiable during backpropagation. To address this issue, popular straight through estimator (STE) method (Bengio et al. [2013]) is widely used, as it skips the gradient of the threshold function during backpropagation. Such process may create to a gap between trainable real-value mask and binary mask. As shown in Figure 3(a), the trainable mask values exhibit certain ratios across the entire range from 0 to 1 after Sigmoid function, which can be inaccurate when further converting to binary mask via a hard threshold function. To better optimize the trainable mask towards binary values, we propose to apply Gumbel-Sigmoid function to learn the binary mask.

The Gumbel distribution is used to model the extreme value distribution and generate samples from the categorical distribution (Gumbel [1954]). This property is then utilized to create the Gumbel-Softmax (Jang et al. [2016]), a differentiable categorical distribution sampling function. The sample of one category is given by:

$$y_i = \frac{\exp((\log(\pi_i) + g_i)/\tau)}{\sum_{j=1}^{k} \exp((\log(\pi_j) + g_j)/\tau)} \tag{6}$$

Where $\tau$ is the input adjustment parameter, $g_i$ is sample from Gumbel distribution. Inspired by the Gumbel-Softmax, we treat learning the binary mask of each point as a two-class category problem. Thus, we replace the Softmax function to Sigmoid function, referring to Gumbel-Sigmoid:

$$gs(m) = \frac{exp((\log(m) + g_0)/\tau)}{exp((\log(m) + g_0)/T) + exp(g_1/\tau)} = \frac{1}{1 + exp(-(\log(m) + g_0 - g_1)/\tau)} \tag{7}$$

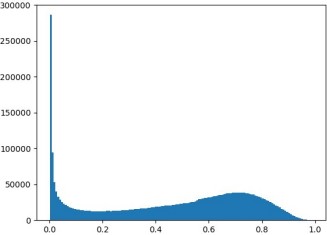
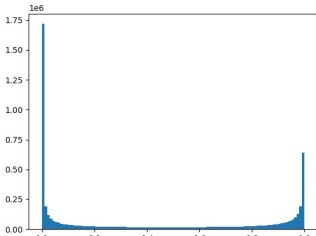

(a) Mask values after Sigmoid activation    (b) Mask values after Gumbel-Sigmoid activation

Figure 3: Comparison between Sigmoid and Gumbel-Sigmoid. The Gumbel-Sigmoid function pushes the values closer to 0 or 1 and is a good approximation of a binarized mask.

By using such Gumbel-Sigmoid function, the output value is either close to 0 or 1, as shown in Figure 3, allowing it to be integrated as an approximation of a binary masking function. More importantly, this function remains differentiable, thus can be integrated during backpropagation.

Moreover, to prune the selected Guassians practically according to the learned binary mask, we further apply the mask value on opacity, which can be mathematically formulated as

$$o_{im} = o_i * gs(m_i * S_i) \tag{8}$$

where $S_i$ is the defined importance score of each Gaussian point. The closer the mask value is to 0, the less corresponding Gaussian point contributes to the rendering. In practice, after learning the trainable mask, a one-time pruning is applied to the corresponding Gaussian points with mask value of 0.

**Sparsity regularization** In order to compress the model as much as possible, we apply a L1 regularization term (Lee et al. [2023]) to encourage the trainable mask to be sparse, which can be formulated as:

$$R_{mask} = \frac{1}{N} \sum_{i=1}^{N} |m_i| \tag{9}$$

Upon that, the final loss function is defined as:

$$L = (1 - \lambda_{ssim}) * L_{L1} + \lambda_{ssim} * L_{ssim} + \lambda_m * R_{mask} \tag{10}$$

$L_{L1}$ is the L1 loss between rendered image and ground truth. $L_{ssim}$ is the ssim loss. $\lambda_{ssim}$ and $\lambda_m$ are two coefficients.

Moreover, we find that the trainable mask can be effectively learned in just a few hundred iterations, compared to the thousands required for the overall training process. In practice, the mask learning function is activated for only 500 iterations. Once the mask values are learned, we follow the 3DGS training setup to further fine-tune the pruned model while maintaining the same total number of training iterations. The detailed hyper parameters are described in the later experiment section.

# 4 Experiments

## 4.1 Experimental Settings

**Dataset and Baseline**  We test our method on two of the most popular real-world datasets: Mip-NeRF360 dataset (Barron et al. [2022]), which contains 9 scenes, and the *Train* and *Truck* scenes from the Tanks & Temples dataset (Knapitsch et al. [2017]). We also evaluate our method on the NeRF-Synthetic dataset (Mildenhall et al. [2021]), which includes 8 synthetic scenes. In this section, we only present the results on MipNeRF360 dataset, rest of them are listed in appendix A. In this paper, we use the SoTA RadSplat (Niemeyer et al. [2024]) and Mini-Splatting (Fang and Wang [2024]) as the baselines, both of which propose different pruning importance scores. First, we evaluate the performance of these two methods under various pruning ratios. Since neither method is open-sourced at the time of writing, we reproduced them based on the provided equations. For each pruning ratio, we calculate the corresponding threshold based on the magnitude of the importance scores and prune the Gaussians with scores below this threshold. Note that each pruning ratio requires one round of training. We use peak signal-to-noise ratio (PSNR), structural similarity index measure (SSIM), and Learned Perceptual Image Patch Similarity (LPIPS) (Zhang et al. [2018]) as rendering evaluation metrics.

**Implement Details**  The machine running the experiments is equipped with an AMD 5955WX processor and two Nvidia A6000 GPUs. It should be noted that our method does not support multi-GPU training. We ran different experiments simultaneously on two GPUs. We train each scene under every setting for 30,000 iterations and with the mask training occurring from iteration 19,500 to 20,000, updating the importance score every 20 iterations. The value of $\tau$ in Equation 7 is 0.5 and the coefficient $\lambda_m$ of mask loss is 5e-4.

## 4.2 Experimental Results

**Quantitative Results**  The blue lines in Figure 4 show the results of sweeping pruning ratios using RadSplat and Mini-Splatting for the *Kitchen* and *Room* scenes. Results for other scenes in the MipNeRF 360 dataset are presented in Appendix A. The result of the learned LP-3DGS model size and rendering quality is indicated by red triangles.

The quantitative results fluctuate at lower pruning ratios but generally stabilize around a certain value. After surpassing that point, the rendering quality decreases significantly. It's worth noting that this critical point varies for different scenes. Rather than manually searching for the favorable pruning ratio, it clearly shows our LP-3DGS method could learn the favorable model size in conjunction with the scene learning process, with only one-time training. The Table 1 lists the quantitative results of all scenes in MipNeRF360 dataset. It clearly shows that each scene converge into different model size leveraging our LP-3DGS method while maintaining almost the same rendering quality. The pruning ratio varies based on what importance score is used, but LP-3DGS effectively identifies the ideal pruning ratio for the corresponding score.

| Scene | Bicycle | Bonsai | Counter | Kitchen | Room | Stump | Garden | Flowers | Treehill | AVG |
|---|---|---|---|---|---|---|---|---|---|---|
| Baseline PSNR ↑ | 25.087 | 32.262 | 29.079 | 31.581 | 31.500 | 26.655 | 27.254 | 21.348 | 22.561 | 27.48 |
| LP-3DGS (RadSplat Score) | 25.099 | 32.094 | 28.936 | 31.515 | 31.490 | 26.687 | 27.290 | 21.383 | 22.706 | 27.47 |
| LP-3DGS (Mini-Splatting Score) | 24.906 | 31.370 | 28.4098 | 30.785 | 31.132 | 26.679 | 27.095 | 21.150 | 22.522 | 27.12 |
| Baseline SSIM ↑ | 0.7464 | 0.9460 | 0.9138 | 0.9320 | 0.9249 | 0.7700 | 0.8557 | 0.5876 | 0.6358 | 0.8125 |
| LP-3DGS (RadSplat Score) | 0.7458 | 0.9441 | 0.9120 | 0.9311 | 0.9243 | 0.7714 | 0.8548 | 0.5865 | 0.6381 | 0.8120 |
| LP-3DGS (Mini-Splatting Score) | 0.7373 | 0.9358 | 0.9017 | 0.9249 | 0.9167 | 0.7677 | 0.8493 | 0.5756 | 0.6336 | 0.8047 |
| Baseline LPIPS ↓ | 0.2441 | 0.1799 | 0.1839 | 0.1164 | 0.1978 | 0.2423 | 0.1224 | 0.3601 | 0.3469 | 0.2215 |
| LP-3DGS (RadSplat Score) | 0.2516 | 0.1865 | 0.1896 | 0.1194 | 0.2032 | 0.2466 | 0.1270 | 0.3656 | 0.3527 | 0.2269 |
| LP-3DGS (Mini-Splatting Score) | 0.2642 | 0.2036 | 0.2068 | 0.1292 | 0.2208 | 0.2553 | 0.1353 | 0.3753 | 0.3618 | 0.2391 |
| RadSplat Score pruning ratio | 0.64 | 0.65 | 0.66 | 0.58 | 0.74 | 0.65 | 0.59 | 0.59 | 0.59 | 0.63 |
| Mini-Splatting Score pruning ratio | 0.57 | 0.67 | 0.64 | 0.56 | 0.71 | 0.61 | 0.60 | 0.54 | 0.54 | 0.60 |

Table 1: The results comparison on the MipNeRF360 dataset shows that LP-3DGS has similar performance after pruning and achieves different pruning ratios for different scenes. This demonstrates LP-3DGS's ability to adaptively find the favorable pruning ratio, maintaining performance while effectively compressing the model.

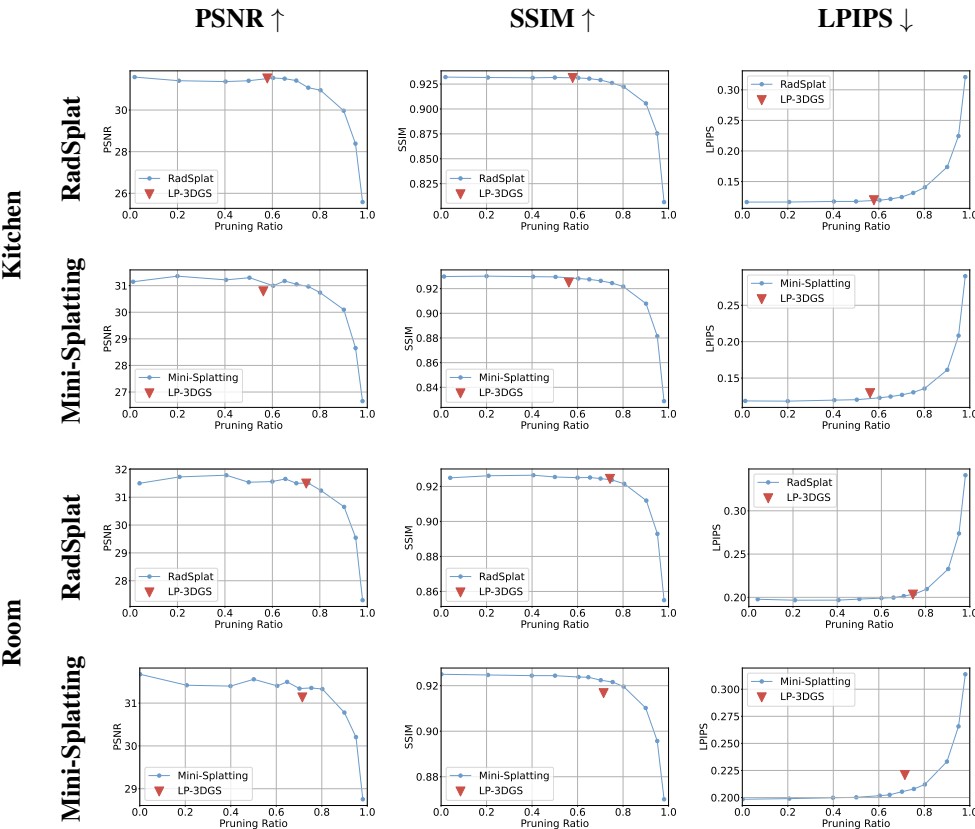

Figure 4: The performance changes with the pruning ratio in different scenes

**Training Cost**    Table 2 shows the training cost of LP-3DGS on MipNeRF360 dataset. In our setup, after 20000th iteration, the model is pruned based on the learned mask values. The number of Gaussian points will be significantly reduced, resulting in the later stages of training taking much less time than in the non-pruned version. Even with the embedding of the mask learning function, the overall training cost is comparable to that of the vanilla 3DGS. In most cases, the peak training memory usage is slightly higher because training the mask requires more GPU memory. However, after pruning, the 3DGS model size becomes much smaller, leading to a significant improvement in rendering speed, measured in terms of FPS.

| Scene | Bicycle | Bonsai | Counter | Kitchen | Room | Stump | Garden | Flowers | Treehill |
|---|---|---|---|---|---|---|---|---|---|
| 3DGS Training time (Minute) | 49 | 34 | 26 | 33 | 27 | 37 | 47 | 33 | 32 |
| LP-3DGS (RadSplat Score) | 43 | 27 | 28 | 34 | 30 | 35 | 46 | 34 | 33 |
| LP-3DGS (Mini-Splatting Score) | 44 | 27 | 28 | 35 | 29 | 34 | 46 | 34 | 33 |
| 3DGS Peak Memory (GB) | 14.7 | 8.6 | 9.4 | 9.3 | 10.6 | 12.2 | 15.7 | 10.3 | 9.4 |
| LP-3DGS (RadSplat Score) | 16.1 | 8.5 | 11.3 | 12.3 | 11.8 | 12.1 | 15.8 | 10.1 | 10.3 |
| LP-3DGS (Mini-Splatting Score) | 15.5 | 8.4 | 12.7 | 13.0 | 13.0 | 12.1 | 15.2 | 9.7 | 9.7 |
| 3DGS FPS | 132 | 417 | 421 | 315 | 380 | 164 | 129 | 200 | 205 |
| LP-3DGS (RadSplat Score) | 324 | 662 | 670 | 542 | 692 | 371 | 296 | 412 | 411 |
| LP-3DGS (Mini-Splatting Score) | 290 | 634 | 650 | 507 | 662 | 341 | 252 | 368 | 384 |

Table 2: Training cost and on MipNeRF360 Dataset. Training time of LP-3DGS is similar with baseline but since the model is compressed, the FPS is larger.

## 4.3   Ablation Study

A recent prior work, Compact3D (Lee et al. [2023]) proposes to leveraging the straight-through estimator (STE) to train a binary mask based on the opacity and scale of Gaussian parameters. To conduct a fair comparison between STE based mask and our LP-3DGS, we perform two ablation

studies, one involves replacing the STE mask in Compact3D with our method, and the other applies the STE mask to the importance score of RadSplat. The formula of STE mask is

$$M(m) = \overline{\nabla}(\mathbb{1}[f(m) > \epsilon] - f(m)) + f(m) \tag{11}$$

$\overline{\nabla}$ means stop gradients, $\mathbb{1}[\cdot]$ is the indicator function and $f(\cdot)$ is sigmoid function, $\epsilon$ is masking threshold.

**Comparison with Compact3D**   We first apply the Gumbel-sidmoid activated mask, instead of STE mask, on the opacity and scale of gaussians in the same manner as proposed in Compact3D. The threshold $\epsilon$ in Equation 11 and mask loss coefficient follows the default settings in Compact3D. Table 3 shows the comparison between two methods.

| | Scene | Bicycle | Bonsai | Counter | Kitchen | Room | Stump | Garden | Flowers | Treehill | AVG |
|---|---|---|---|---|---|---|---|---|---|---|---|
| PSNR | Compact3D | 24.846 | 32.19 | 29.066 | 30.867 | 31.489 | 26.408 | 27.026 | 21.187 | 22.479 | 27.284 |
| | LP-3DGS | 25.087 | 32.2 | 29.033 | 31.213 | 31.678 | 26.658 | 27.223 | 21.32 | 22.569 | 27.442 |
| SSIM | Compact3D | 0.7292 | 0.9462 | 0.9137 | 0.925 | 0.9251 | 0.7563 | 0.8446 | 0.5773 | 0.6305 | 0.8053 |
| | LP-3DGS | 0.7438 | 0.9461 | 0.9141 | 0.9305 | 0.9263 | 0.7687 | 0.8547 | 0.5843 | 0.6358 | 0.8116 |
| LPIPS | Compact3D | 0.266 | 0.1815 | 0.1866 | 0.124 | 0.2012 | 0.2615 | 0.1401 | 0.3722 | 0.3555 | 0.2320 |
| | LP-3DGS | 0.2526 | 0.1833 | 0.1867 | 0.1201 | 0.2013 | 0.2472 | 0.1275 | 0.3668 | 0.3513 | 0.2263 |
| #Gaussians | Compact3D | 2620663 | 666558 | 570126 | 1050079 | 566332 | 1902711 | 2412796 | 1685224 | 2089515 | 1507109 |
| | LP-3DGS | 2510992 | 542235 | 506391 | 887161 | 479681 | 2014270 | 2836989 | 1747766 | 1804155 | 1481071 |

Table 3: Results with/without trainable mask on Gaussian opacity and scale

In most cases, our LP-3DGS learns a higher pruning ratio, except for Stump, Garden and Flowers scene. In terms of rendering quality, our LP-3DGS outperforms Compact3D using the STE based mask achieving even smaller model sizes in most scenes.

**STE Mask on Importance Score**   We also apply STE mask on the pruning importance score to compare with our method. The Equation 8 would be rewriten as

$$o_{im} = o_i * M(m_i * is) \tag{12}$$

where $M$ is shown in Equation 11. The same as mentioned before, the parameters for STE mask are default values in Compact3D.

| | Scene | Bicycle | Bonsai | Counter | Kitchen | Room | Stump | Garden | Flowers | Treehill | AVG |
|---|---|---|---|---|---|---|---|---|---|---|---|
| PSNR | LP-3DGS | 25.099 | 32.094 | 28.936 | 31.515 | 31.490 | 26.687 | 27.290 | 21.383 | 22.706 | 27.470 |
| | STE mask | 24.833 | 30.947 | 28.371 | 30.705 | 30.950 | 26.396 | 26.793 | 21.056 | 22.552 | 26.955 |
| SSIM | LP-3DGS | 0.7458 | 0.9441 | 0.9120 | 0.9311 | 0.9243 | 0.7714 | 0.8548 | 0.5865 | 0.6381 | 0.8120 |
| | STE mask | 0.7231 | 0.9268 | 0.8925 | 0.9162 | 0.9120 | 0.7514 | 0.8289 | 0.5624 | 0.6196 | 0.7922 |
| LPIPS | LP-3DGS | 0.2441 | 0.1799 | 0.1839 | 0.1164 | 0.1978 | 0.2423 | 0.1224 | 0.3601 | 0.3469 | 0.2215 |
| | STE mask | 0.2937 | 0.2194 | 0.2274 | 0.1480 | 0.2334 | 0.2899 | 0.1771 | 0.3988 | 0.3983 | 0.2651 |
| Pruning Ratio | LP-3DGS | 0.64 | 0.65 | 0.66 | 0.58 | 0.74 | 0.65 | 0.59 | 0.59 | 0.59 | 0.63 |
| | STE mask | 0.84 | 0.88 | 0.88 | 0.87 | 0.89 | 0.86 | 0.85 | 0.83 | 0.83 | 0.86 |

Table 4: Results using LP-3DGS and STE mask on importance score of RadSplat

Table 4 shows that under the same settings, after applying the mask to the importance score, the STE mask compresses the mode too much and the performance drops a lot. Trainable mask keeps the gradient of the mask and the comressed model has a more reasonable size.

| | Scene | Bicycle | Bonsai | Counter | Kitchen | Room | Stump | Garden | Flowers | Treehill | AVG |
|---|---|---|---|---|---|---|---|---|---|---|---|
| PSNR | LP-3DGS | 24.906 | 31.370 | 28.4098 | 30.785 | 31.132 | 26.679 | 27.095 | 21.150 | 22.522 | 27.12 |
| | STE mask | 24.894 | 30.925 | 28.334 | 30.731 | 31.032 | 26.470 | 26.863 | 20.997 | 22.559 | 26.98 |
| SSIM | LP-3DGS | 0.7373 | 0.9358 | 0.9017 | 0.9249 | 0.9167 | 0.7677 | 0.8493 | 0.5756 | 0.6336 | 0.8047 |
| | STE mask | 0.7287 | 0.9292 | 0.8978 | 0.9232 | 0.9152 | 0.7562 | 0.8381 | 0.5629 | 0.6247 | 0.7973 |
| LPIPS | LP-3DGS | 0.2642 | 0.2036 | 0.2068 | 0.1292 | 0.2208 | 0.2553 | 0.1353 | 0.3753 | 0.3618 | 0.2391 |
| | STE mask | 0.2821 | 0.2154 | 0.2145 | 0.1330 | 0.2241 | 0.2797 | 0.1564 | 0.3939 | 0.3852 | 0.2538 |
| Pruning Ratio | LP-3DGS | 0.57 | 0.67 | 0.64 | 0.56 | 0.71 | 0.61 | 0.60 | 0.54 | 0.54 | 0.60 |
| | STE mask | 0.75 | 0.77 | 0.75 | 0.66 | 0.79 | 0.80 | 0.75 | 0.75 | 0.75 | 0.75 |

Table 5: Results using LP-3DGS and STE mask on importance score of Mini-Splatting

# 5   Discussion and Conclusion

**Broader Impact and Limitation**   LP-3DGS compresses the 3DGS model to an ideal size in a single run, saving storage and computational resources by eliminating the need for parameter sweeping to find the favorable pruning ratio. However, the limitation of this work is that the rendering quality after pruning varies depending on the the definition of importance scores.

**Conclusion**   In this paper, we present a novel framework, LP-3DGS, which guides the 3DGS model learn the best model size. The framework applies a trainable mask on the importance score of the gaussian points. The mask is trained for a specific period and used to prune the model once. Our method compressed the model as much as possible without significantly sacrificing performance and is capable to achieve the favorable compression rate for different test scenes. Compared with the STE mask method, ours achieves better performance.

## Acknowledgments and Disclosure of Funding

Supported by the Intelligence Advanced Research Projects Activity (IARPA) via Department of Interior/ Interior Business Center (DOI/IBC) contract number 140D0423C0076. The U.S. Government is authorized to reproduce and distribute reprints for Governmental purposes notwithstanding any copyright annotation thereon. Disclaimer: The views and conclusions contained herein are those of the authors and should not be interpreted as necessarily representing the official policies or endorsements, either expressed or implied, of IARPA, DOI/IBC, or the U.S. Government.

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

# A Appendix / supplemental material

*Code:* https://github.com/dexgfsdfdsg/LP-3DGS.git

## A.1 Experiment Results on MipNeRF360 Dataset

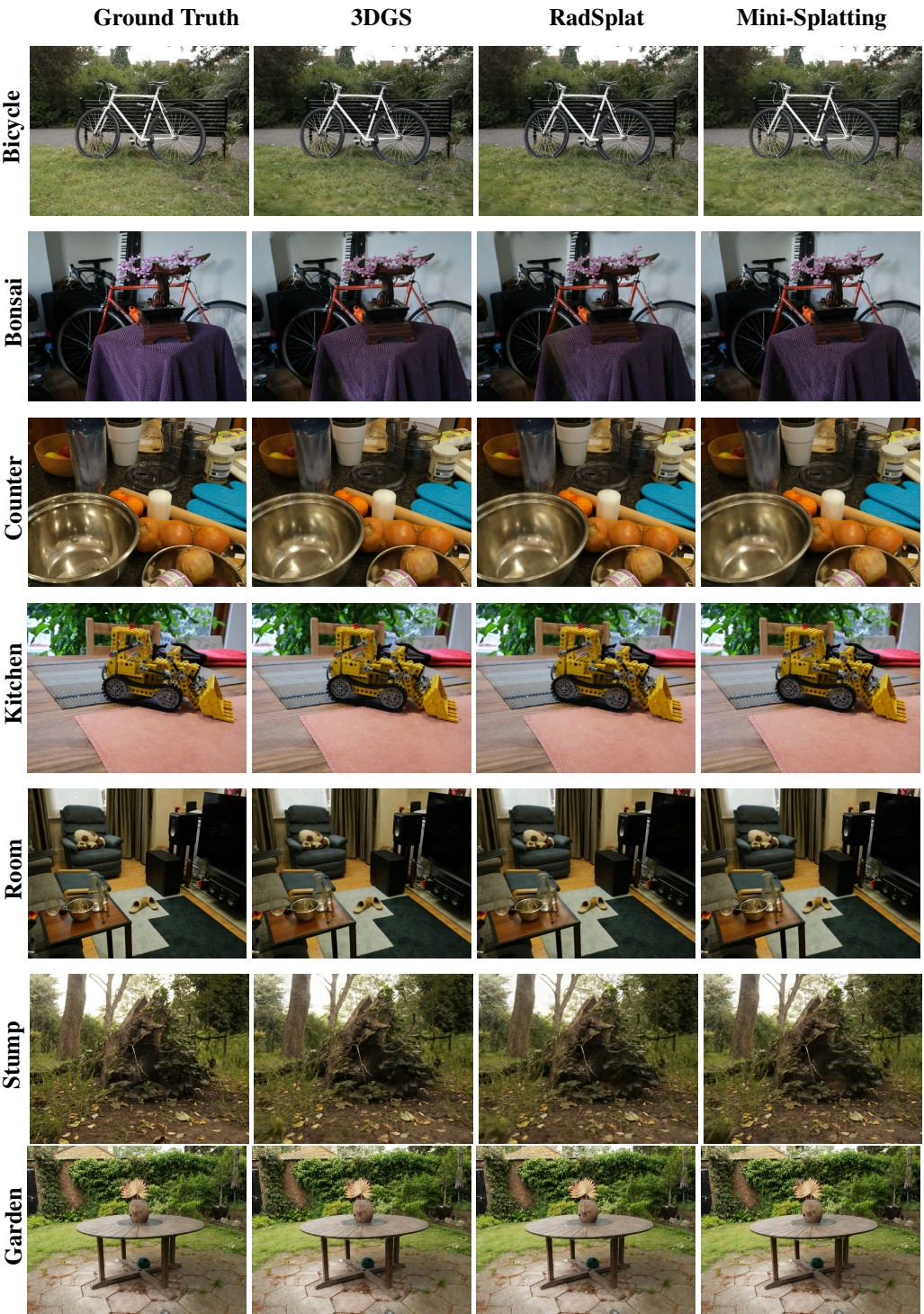

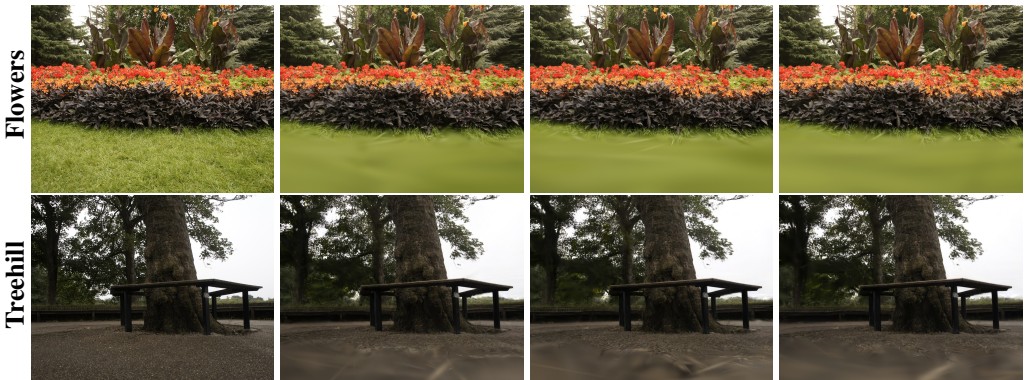

Figure 5: Rendered images on MipNeRF360 Dataset

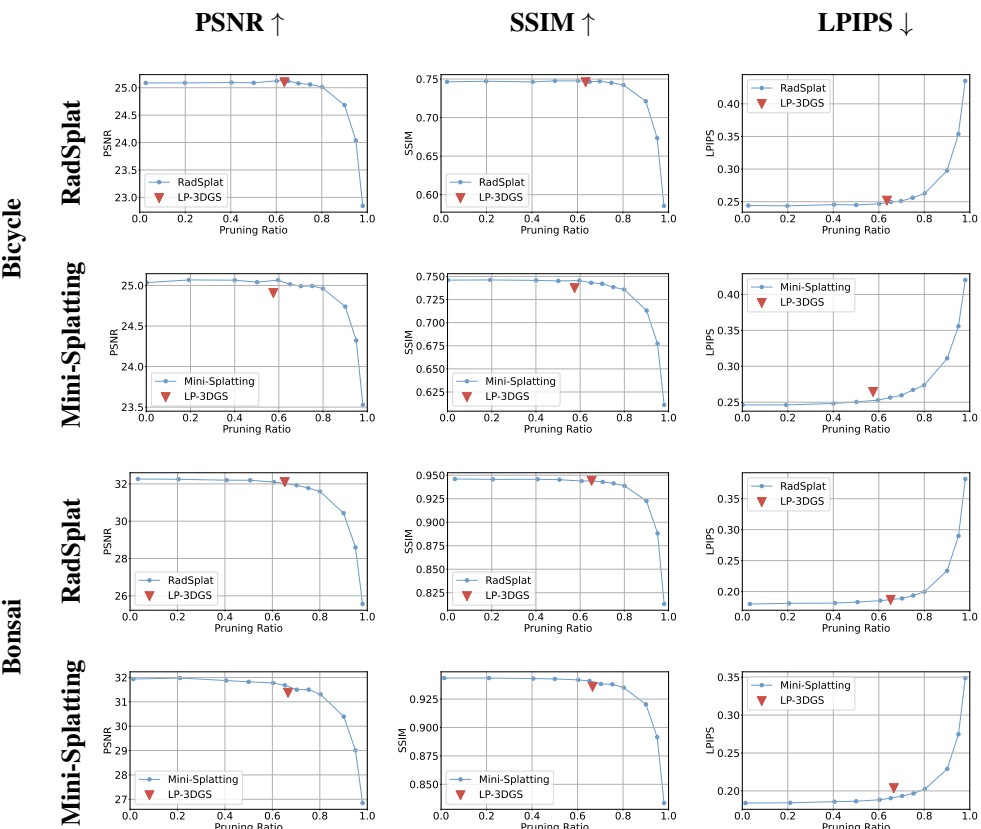

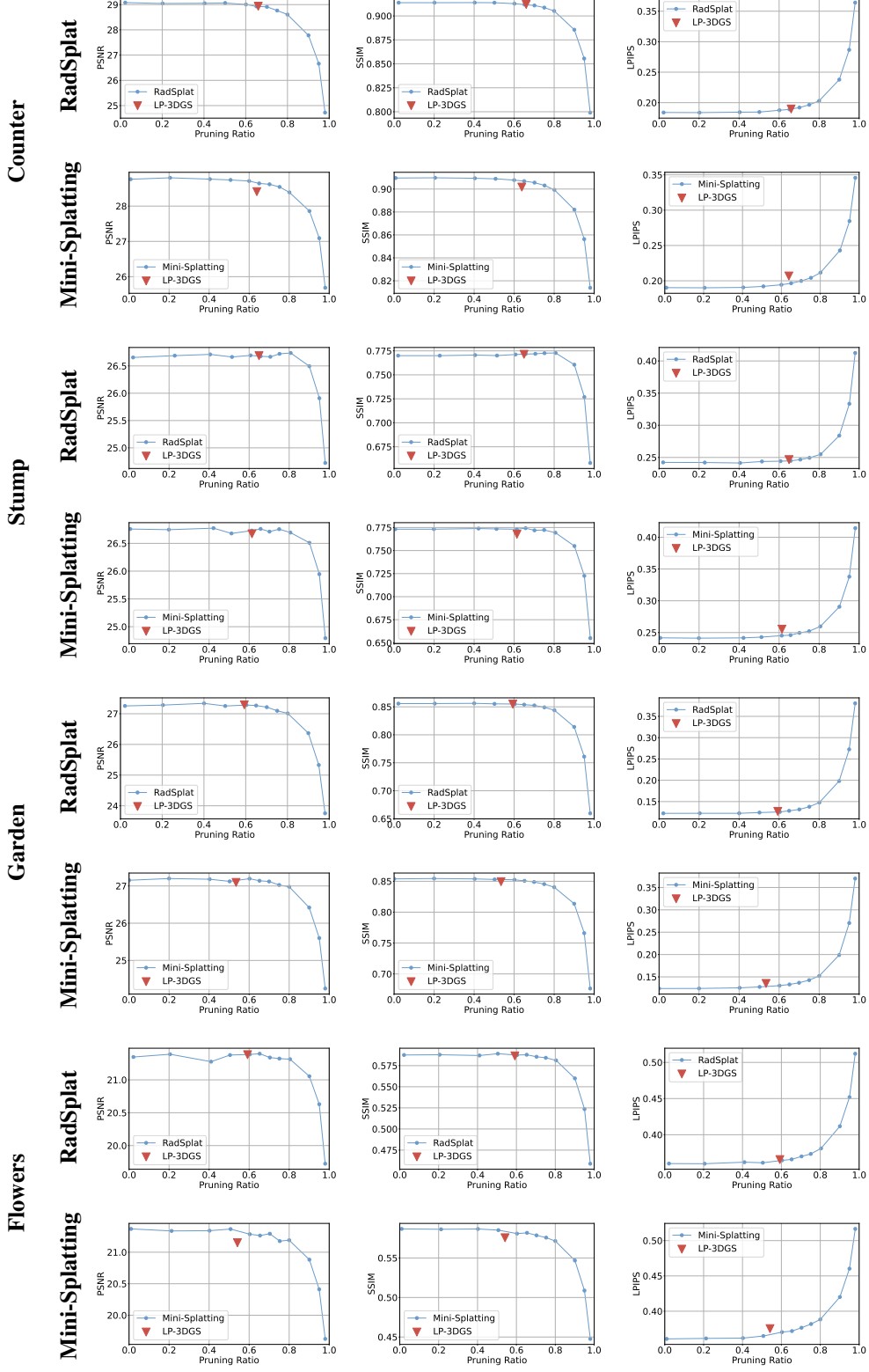

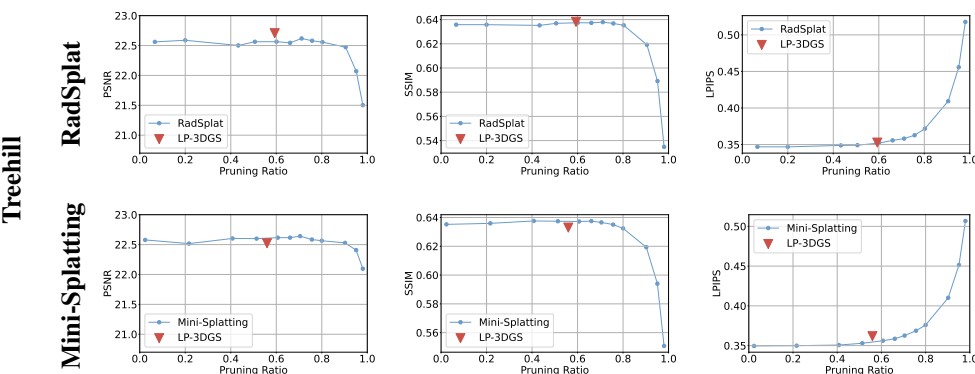

Figure 6: The performance changes with the pruning ratio in different scenes

## A.2 Experiment Results on NeRF Synthetic Dataset

| Scene | Chair | Drums | Ficus | Hotdog | Lego | Materials | Mic | Ship | AVG |
|---|---|---|---|---|---|---|---|---|---|
| Baseline PSNR ↑ | 35.546 | 26.276 | 35.480 | 38.081 | 36.012 | 30.502 | 36.795 | 31.688 | 33.798 |
| LP-3DGS (RadSplat Score) | 35.496 | 26.221 | 35.442 | 37.976 | 35.990 | 30.374 | 36.589 | 31.584 | 33.709 |
| LP-3DGS (Mini-Splatting Score) | 35.419 | 26.102 | 35.354 | 37.728 | 35.769 | 29.883 | 36.337 | 31.375 | 33.496 |
| Baseline SSIM ↑ | 0.9877 | 0.9548 | 0.9870 | 0.9854 | 0.9825 | 0.9604 | 0.9926 | 0.9062 | 0.9696 |
| LP-3DGS (RadSplat Score) | 0.9878 | 0.9547 | 0.9867 | 0.9854 | 0.9825 | 0.9598 | 0.9924 | 0.9061 | 0.9694 |
| LP-3DGS (Mini-Splatting Score) | 0.9874 | 0.9358 | 0.9867 | 0.9846 | 0.9817 | 0.9566 | 0.9919 | 0.9034 | 0.966 |
| Baseline LPIPS ↓ | 0.01046 | 0.03657 | 0.01775 | 0.01977 | 0.0161 | 0.03671 | 0.00635 | 0.1058 | 0.03119 |
| LP-3DGS (RadSplat Score) | 0.01091 | 0.03723 | 0.01213 | 0.02079 | 0.01675 | 0.03817 | 0.00680 | 0.1083 | 0.03139 |
| LP-3DGS (Mini-Splatting Score) | 0.0111 | 0.03876 | 0.01217 | 0.02211 | 0.018 | 0.04323 | 0.00749 | 0.1151 | 0.0335 |
| RadSplat Score pruning ratio | 0.77 | 0.76 | 0.84 | 0.68 | 0.65 | 0.61 | 0.78 | 0.60 | 0.71 |
| Mini-Splatting Score pruning ratio | 0.63 | 0.65 | 0.65 | 0.58 | 0.58 | 0.80 | 0.60 | 0.50 | 0.62 |

Table 6: The quantitative results on NeRF Synthetic Dataset

## A.3 Experiment Results on Truck & Train Scenes

| Scene | Truck | Train | AVG |
|---|---|---|---|
| Baseline PSNR ↑ | 25.263 | 22.025 | 23.644 |
| LP-3DGS (RadSplat Score) | 25.376 | 21.822 | 23.599 |
| LP-3DGS (Mini-Splatting Score) | 25.152 | 21.675 | 23.414 |
| Baseline SSIM ↑ | 0.8778 | 0.8118 | 0.8448 |
| LP-3DGS (RadSplat Score) | 0.8768 | 0.8072 | 0.8420 |
| LP-3DGS (Mini-Splatting Score) | 0.8724 | 0.7963 | 0.8344 |
| Baseline LPIPS ↓ | 0.1482 | 0.2083 | 0.1783 |
| LP-3DGS (RadSplat Score) | 0.1541 | 0.2217 | 0.1879 |
| LP-3DGS (Mini-Splatting Score) | 0.162 | 0.2343 | 0.1982 |
| RadSplat Score pruning ratio | 0.72 | 0.63 | 0.68 |
| Mini-Splatting Score pruning ratio | 0.65 | 0.57 | 0.61 |

Table 7: The quantitative results on Tanks & Temples Dataset

