# OpenReview forum: "LP-3DGS: Learning to Prune 3D Gaussian Splatting"
_NeurIPS.cc/2024/Conference — NeurIPS 2024 poster_

### Official Review · Reviewer_1UP2 · 2024-07-10

**Soundness:** 2
**Presentation:** 2
**Contribution:** 2
**Rating:** 5
**Confidence:** 4

**Summary:**

The paper proposed a pruning method for Gaussian Splatting training, which employ a learnable mask to find optimal pruning rate.

**Strengths:**

The paper is well-written and the proposed method is easy to understand.

**Weaknesses:**

* The proposed method use a regularization loss to encourage the model to prune, so that the weight parameter is an important balancing factor. The discussion on how to set such parameter is missing.

* Figure 2 is not clear enough for me to well understand the proposed idea. In Figure 3, why pattern like (b) is better than (a)?

* Experiments section use existing works for comparison, but they are all self-implemented. The performance in their original paper should be included for reference.

* The author claim the proposed method can find optimal pruning ratio. Based on my observation, the conclusion is not solid. First, what is definition of optimal? Second, not all the subfigures in Fig. 4 & 6 support this conclusion.

**Questions:**

Please see the "weaknesses".

**Limitations:**

No potential negative social impact.

---

> ### Author Rebuttal · Authors · 2024-08-07
>
> ### Response to weakness 1:
>
> We would like to emphasize that the main contribution of our work is not merely the use of a regularization loss to encourage the model to prune, but rather the development of a learning framework that can automatically learn a pruning ratio embedding during the training process of a 3DGS model. This is achieved through our proposed Gumbel-Sigmoid mask pruning technique (Equation 8) and our ensemble loss function (Equation 10), all with only one round of training. We kindly request that the reviewer refer to our summary of technical contributions in the Common Question.
>
> For the setting of $\lambda_{m}$, we used a value of 5e-4 in all the reported experimental results. We have further conducted an ablation study on this hyperparameter based on the Room scene, as detailed below.
>
> | $\lambda_{m}$ | PSNR  | SSIM  | LPIPS | Pruning Ratio |
> |---------------|-------|-------|-------|---------------|
> | 3e-4          | 31.488| 0.9241| 0.2035| 73.15%        |
> | 5e-4          | 31.490| 0.9243| 0.2032| 73.78%        |
> | 7e-4          | 31.474| 0.9238| 0.2040| 75.91%        |
>
> $\lambda_{m}$ have little impact on the final rendering quality.
>
> ### Response to weakness 2:
>
> Figure 2 is based on the vanilla 3DGS, with our proposed pruning method highlighted within the green box. This box indicates that a mask is learned for each Gaussian to determine whether it should be pruned during the training process. The detailed technique described mathematically relates to Equations 7, 8, and 10.
>
> For Figure 3, since the pruning process involves a binary decision (i.e., whether to prune or not prune), it is preferable for the learned mask distribution to naturally reflect a binary distribution—where values close to 0 indicate the Gaussian should be pruned, and values close to 1 indicate it should not be pruned.
>
> ### Response to weakness 3:
>
> Since we need to report the training time for an apples-to-apples comparison, it is essential to use the same GPU hardware setup. We have open-sourced all our code and experimental results on GitHub for peer researchers to verify and benchmark.
>
> For 3DGS and Mini-Splatting, we reported results using their open-source codes on our GPU hardware. Additionally, because RadSplat is not open-sourced, we re-implemented and benchmarked their method ourselves. It is important to note that all our self-implemented results are consistent with the rendering quality reported in the original papers. To further clarify, we have provided a comparison of the original paper results for 3DGS and our self-implemented results as follows:
>
> | Scene           | Bicycle | Bonsai | Counter | Kitchen | Room | Stump | Garden | Flowers | Treehill | **AVG** |
> |-----------------|---------|--------|---------|---------|------|-------|--------|---------|----------|---------|
> | **Original paper PSNR** | 25.246  | 31.980 | 28.700  | 30.317  | 30.632 | 26.550 | 27.410 | 21.520  | 22.490  | 27.205  |
> | **Self-Implemented PSNR**   | 25.087  | 32.262 | 29.079  | 31.581  | 31.500 | 26.655 | 27.254 | 21.348  | 22.561  | 27.480  |
> ||
> | **Original paper SSIM** | 0.771   | 0.938  | 0.905   | 0.922   | 0.914 | 0.775 | 0.868  | 0.605   | 0.638   | 0.8151  |
> | **Self-Implemented SSIM**   | 0.7464  | 0.9460 | 0.9138  | 0.9320  | 0.9249 | 0.770 | 0.8557 | 0.5876  | 0.6358  | 0.8125  |
> ||
> | **Original paper LPIPS** | 0.205   | 0.205  | 0.204   | 0.129   | 0.220 | 0.210 | 0.103  | 0.336   | 0.317   | 0.2143  |
> | **Self-Implemented LPIPS**   | 0.2441  | 0.1799 | 0.1839  | 0.1164  | 0.1978 | 0.2423 | 0.1224 | 0.3601  | 0.3469  | 0.2215  |
>
>
> ### Response to weakness 4:
>
> Please refer to the Common Question about the term "optimal" pruning for additional details.

---

> > ### Comment · Reviewer_1UP2 · 2024-08-12
> >
> > While the authors addressed the most of my minor concerns, I still insist on my opinion about the key point about the "optimal prune ratio".
> >
> > First, according to the experiments, most of the "optimal" prune ratios are similar, around 0.7-0.8 without specific pattern.  It is a very narrow band. In such cases, the "learning" task for the prune ratio may be a trivial task since the prune ratio is easy to approach. Just set a 0.75 value may be good enough for most of the experiments in Fig.4 & 6.
> >
> > Second, even though we assume the task is non-trivial, the experiment cannot prove the proposed strategy is efficient enough. For example, in the Bicycle-Redsplat subfigures of  Fig.6, we cannot tell the prune ratio ~0.6(LP-3DGS) is better than 0.7 or 0.8. Most of the experiments are with the same problem: no convincing metric to examine the efficiency of the proposed method. Actually, I think it is the problem of this task as I said in the first paragraph.
> >
> > In other words, if all the "answers" fall in around 0.6-0.8, and we cannot find the concrete value which is significantly better than others, the results are too vague to support the claims.
> >
> > Therefore for now I will keep my score.

---

> ### Author Response · Authors · 2024-08-11
>
> Dear reviewer,
> We hope our rebuttal can address your comments and concern. As the deadline for the author-reviewer discussion is approaching, would you please share your thoughts on our rebuttal and reconsider the review score? thanks a lot

---

> > ### Author Response · Authors · 2024-08-13
> >
> > Thanks for your comments
> >
> > ### For "optimal" wording
> > As for the term of "optimal pruning ratio", in our response to the common question, we already claimed that we will not use it and change to "learned pruning ratio" with more detailed definition. Please refer to above for our detailed clarification. The reviewer o7SR suggested us to change to "learned effective pruning ratio" and we plan to use such in our updated manuscript.
> >  For the sensitivity analysis of rendering image quality against pruning ratios, as we show in our experiment results (figure 4 and 6), each scene has different sensitivities against pruning and the "band" is in general within around 0.5 to 0.8, rather than 0.7 to 0.8 or 0.6 to 0.8, where the rendering image quality starts to drop with increasing pruning ratio. In order to find such "band" for different scenes, a manual tuning of pruning ratios with multiple rounds of training is still needed in prior works, which is costly and not an "easy approach". We kindly disagree that just setting the pruning ratio to 0.75 will be just fine.
> > To justify such, we conducted extra experiments based on the pruning ratio of 0.75 based on the Radsplat score for kitchen and room scenes, due to the limited rebuttal time left. Note that, table 1 in our manuscript reported the complete set of our method learned pruning ratio and its corresponding image quality of each scene. From table 1, it can be seen that, in average, our model with learned pruning ratio has almost the same rendering image quality as the unpruned version, which is what we defined in the loss function and the experiment results support that well. In particular, the learned pruning ratio is 0.58 for kitchen scene and 0.74 for room scene. It can be seen from the extra experiments in below table that the kitchen scene has a significant quality drop with a random 0.75 pruning ratio than that of learned value of 0.58. While, since the randomly chosen 0.75 pruning ratio is close to the learned ratio value of 0.74 for Room scene, the rendering image quality is very similar.
> > Furthermore, such 0.75 pruning ratio is actually not "random", but from the thorough complete pruning sensitivity analysis reported in our submitted manuscript. We believe these ablation study could further prove the motivation of this work and effectiveness of our proposed method.
> > To summarize, for scientific research and contributions, we believe our work discovered the pattern of pruning sensitivity in 3DGS and further propose a general learning method that could be applied to different types of pruning importance scores with a target of automatic pruning, which is a solid contribution to the community and will inspire more research in the future.
> >
> > |       | Pruning Raio | Kitchen |  Room  |
> > |:-----:|:------------:|:-------:|:------:|
> > |       |       0      |  31.581 | 31.500 |
> > |  PSRN |    Learned   |  31.515 | 31.490 |
> > |       |     0.75     |  31.187 | 31.420 |
> > ||
> > |       |       0      |  0.9230 | 0.9249 |
> > |  SSIM |    Learned   |  0.9311 | 0.9243 |
> > |       |     0.75     |  0.9270 | 0.9235 |
> > ||
> > |       |       0      |  0.1164 | 0.1978 |
> > | LPIPS |    Learned   |  0.1194 | 0.2032 |
> > |       |     0.75     |  0.1293 | 0.2045 |
> >
> > ### We are not sure whether the reviewers are criticising the effectiveness or efficiency. We discussed about both in below to clarify.
> >
> > Method effectiveness: For the Bicycle-Radsplat subfigure of Figure 6, all the quantitative metrics, i.e., PSNR/SSIM/LPIPS, that are very popularly used in this community to evaluate the rendering image quality are starting to become worse beyond 0.6 pruning ratio. It actually shows the effectiveness of our method that could learn such pruning ratio with only one time training. The pruning is always a tradeoff process between compression ratio and rendering quality, which is the nature of this problem. We never claimed our learned pruning ratio is always better than other manual pruning ratio settings, but the key point is our method is learnable and automatic that could efficiently and effectively achieve a pruning ratio that is always within the "band" mentioned by the reviewer, rather than costly manual tuning.
> >
> > Method efficiency: We conducted comprehensive training cost analysis in table 2. The actual training time compared with vanilla 3DGS reduces even for single round of training. Compared with other manual pruning method, as we discussed before, our method could achieve a learned effective pruning ratio, embedding with the 3DGS model construction learning process, without the costly pruning sensitivity analysis shown in the figure 4 and 6 that require multiple rounds of training with preset pruning ratio. We believe our method could improve both the 3DGS learning efficiency and pruning efficiency significantly compared with prior works.
> >
> > We kindly request the reviewer to reconsider our technical contributions to the community, as well as the final review score of our work.

---

> > > ### Comment · Reviewer_1UP2 · 2024-08-13
> > >
> > > Maybe the authors misunderstand my first point. I'm not criticizing the wording. In the comment the authors claimed that "the "band" is in general within around 0.5 to 0.8, rather than 0.7 to 0.8 or 0.6 to 0.8".  I do not think there is a substantial difference between them.
> > >
> > > However, I would like to provide some advice to evaluate your method more scientifically.
> > >
> > > To prove that your method is more effective than randomly selecting the prune ratio in the empirical interval e.g. 0.5-0.8, you can randomly select many points from the interval to get the expectation of these metrics including PSNR/LPIPS/SSIM. Compared with LP-3DGS, if there is a significant improvement in each indicator and scenario, it indicates that your method is effective.  For example, I would expect that LP-3DGS outperforms the random strategy in 80%-90% scenes.  As for now, the evaluation results are still vague.
> > >
> > > As for the presentation, you need to demonstrate the strong correlation between the selected prune ratio and these metrics, in the empirical interval of 0.5-0.8 rather than the 0.0-1.0 shown in your figures. The big interval demonstrated(0.0-1.0) in Fig4&6 contains too many impossible value while overwhelming the effectiveness of the proposed methods.

---

> > > > ### Author Response · Authors · 2024-08-13
> > > >
> > > > Dear reviewer,
> > > >
> > > > 1. We did not misunderstand your first comment. In the previous response, we just used one sentence to clarify the word of "optimal". While, the majority rest is about the band interval discussion and clarification of our method. please re-evaluate.
> > > >
> > > > 2. In our reported experiments and our several rounds of discussion above, we already stated very clearly that we never argued our learned pruning ratio is always 'better' than random chosen pruning ratio, since there is no 'better' pruning ratio in model pruning problem. It is always a tradeoff between pruning/compression ratio and quality, which is the nature of this problem. But the key point is our method is learnable and automatic that could efficiently and effectively achieve a pruning ratio that is always within the "band" mentioned by the reviewer, ''with only one round of training and embedding with the 3DGS model learning phase, rather than costly manual tuning''. From this point of view, we strongly believe all the experiment results and extra results provided during this rebuttal period to all reviewers are sufficient and strong enough to prove the motivation and effectiveness of our proposed method.
> > > > We kindly recommend the reviewer to check our discussion with the reviewers o7SR and fvLt above. Both agreed the community will benefit from our proposed automatic pruning and it is more efficient than running entire pruning sensitivity analysis for every scene when compressing the large size of 3DGS model.
> > > >
> > > > 3. As for the suggestions of adding more pruning results during 0.5 to 0.8 interval.  we will start running more experiments as suggested. However, we do not see the point why it is helping for our method, since our method already learned a large pruning ratio that has almost no quality drop for every unique scene, with only one round of training. More points within the 0.5 to 0.8 interval means more training cost for manual pruning ratio tuning, it will only make the benchmarking baseline worse.
> > > >
> > > > 4. Again, we want to highlight that such 0.5 to 0.8 pruning ratio interval "found in this work" is based on the vanilla and default 3DGS training setup. It reveals certain redundancy during the Gaussian densification process. Different 3DGS training hyperparameter settings, for example, the gradient threshold of densification process, will significantly change the number of Gaussian of the final trained model. Then, it is obvious that such "empirical 0.5 to 0.8 interval" will change based on the model redundancy. In this case, even with prior general pruning experience, the cutoff band would change accordingly. To optimizing the tradeoff between compression ratio and rendering quality, new costly pruning sensitivity analysis will be required to get such new cutoff band.
> > > >
> > > > we already open-sourced all our codes and results in the github for the community to peer-review and test on their different pruning importance score.
> > > >
> > > > We appreciate the reviewer's comments and discussion. But for scientific research, we want to make sure the major contribution of this work is highlighted and understood well.

---

> ### Comment · Reviewer_1UP2 · 2024-08-13
>
> Dear authors,
>
> I do agree that "our learned pruning ratio is always 'better' than random chosen pruning ratio, since there is no 'better' pruning ratio in model pruning problem". However, in my previous comment, I only expect that your method is better in over 80% scenes.  I do not require an absolute better or optimal. At least a number larger than 50% can prove that the proposed method is effective.  I do not think a successful rate of 80% is a high standard for a publishable paper. Learnable and automatic method is not preferred if just a random manual number setting can solve the problem comparably.
> This is the most important point in my opinion. If the proposed method can not demonstrate enough effectiveness, I can only regard it as an engineering trick in GS reconstruction.
>
> As for the comment "such 0.5 to 0.8 pruning ratio interval "found in this work" is based on the vanilla and default 3DGS training setup, ... Different 3DGS training hyperparameter settings...". I want to clarify that:
> If you want to prove your method is effective on other variants of GS, please just show evidence.
> If you want to prove that finding empirical prune ratio of other GS is more difficult compared with adopting your method, please show evidence.
>
> Finally, as for your comment "We kindly recommend the reviewer to check our discussion with the reviewers o7SR and fvLt above. Both agreed the community will benefit from our proposed automatic pruning",  I do have seen all the discussions above and **I will keep my independent thinking**.  I've spent lots of time on this paper and I am not an outsider of GS. I kindly recommend the authors repect every review equally no matter they are positive or negative.

---

> > ### Author Response · Authors · 2024-08-13
> >
> > Dear reviewer 1UP2
> >
> > thanks for the comments.
> >
> > 1. If you agree there is no 'better' pruning ratio, but how to guarantee our proposed method could learn the pruning ratio located within the cutoff band in the pruning sensitivity analysis plot shown in figure 4 and 6, what does that mean "At least a number larger than 50% can prove that the proposed method is effective." ? We believe all our learned pruning ratios for every scene is already located within the cutoff band of pruning sensitivity analysis plot. the table 1 also reports, with our learned pruning ratio, almost every scene has the same rendering image quality with the vanilla 3DGS. The average rendering image quality is also the same as the unpruned version with an average of 63% pruning ratio. We do not quite understand what are the requested extra  experiments to prove we are at least 50% better. if you mean rendering quality, the average rendering quality of the pruned model is already the same as unpruned version, but with 2.7X smaller model size (i.e. average 63% pruning ratio)
> >
> > 2. We already setup the experiments with different vanilla 3DGS parameters, which will lead to different initial model redundancy. Hope we could get the results before rebuttal deadline. We will post the results once we get it

---

> > > ### Comment · Reviewer_1UP2 · 2024-08-13
> > >
> > > I understand that you are talking about the trade-off: better compression ratio means lower rendering quality.  However, to demonstrate the proposed method is better than random selection in empirical interval is necessary. The metric that falling into the "band" is so loose.  How about defining a metric for such trade-off task?  Take PSNR as example. In the case of maximum loss of 1dB(example value), the higher the compression rate, the better.

---

> > > > ### Author Response · Authors · 2024-08-13
> > > >
> > > > Dear reviewer 1UP2
> > > >
> > > > thanks for the comments.
> > > >
> > > > We did not only use metric of 'falling into the band' as the evaluation, this is what we want to show the motivation of this work. we actually reported quantitive results of pruning performance in table 1 of our original manuscript. Please refer to that.
> > > > We did not even set 1db drop as the bar, but targeting to achieve almost no rendering quality drop and learn the corresponding pruning ratio. it is proven in the quantitive results in table 1.
> > > >
> > > > For the MipNeRF360 dataset, the average PSNR is 27.48 for unpruned 3DGS model (baseline in table 1). our learned pruned model based on Radsplat score has an average PSNR of 27.47 with average pruning ratio of 0.63. the learned pruned model based on mini-splatting score has an average PSNR of 27.12 with average pruning ratio of 0.6. This difference comes with different definitions of pruning importance scores in both methods, but showing similar performance.
> > > > Also, we want to note that we only use the pruning importance scores defined in Radsplat and Mini-Splatting as the pruning metric from baseline 3DGS model, without any other tricks in those works to further improve the quality, for fair comparison. Therefore, the theoretical performance ceiling is the unpruned model.
> > > >
> > > > For comparing with random pruning ratio selection, the pruning sensitivity analysis in figure 4 and 6 already shows the rendering quality remains almost the same when it is smaller than our learned pruning ratio for each scene. The rendering quality starts to drop when it is larger than our learned pruning ratio.
> > > >
> > > > Hope this is what the reviewer requested for proving the 'effectiveness' of our method.
> > > >
> > > > thanks
> > > >
> > > >
> > > >
> > > >
> > > >
> > > > prune the vanilla

---

> > > > > ### Comment · Reviewer_1UP2 · 2024-08-13
> > > > >
> > > > > "Almost no rendering quality drop" is vague. I recommend you specify the value range and the compression rate in your Tables. The most controversial point we have discussed origins from the task itself. I will raise my score.

---

> > > > > > ### Author Response · Authors · 2024-08-13
> > > > > >
> > > > > > Dear Reviewer 1UP2,
> > > > > >
> > > > > > Thanks for the suggestion and we will update the manuscript as suggested. Appreciate the discussion and consideration for our work

---

### Official Review · Reviewer_CWFJ · 2024-07-13

**Soundness:** 2
**Presentation:** 2
**Contribution:** 1
**Rating:** 4
**Confidence:** 3

**Summary:**

This work introduces LP-3DGS, aiming to compress 3DGS by replacing the previously manually set threshold with a learning-to-prune scheme.  In particular, LP-3DGS learns a binary mask to automatically prune 3DGS, where such a mask is regularized with the Gumbel-Sigmoid technique. Experiments on three benchmarks showcase the soundness of the introduced methods.

**Strengths:**

* Applying Gumbel-Sigmoid to regularize the binary mask is reasonable.
* Extensive experiments are conducted on several existing benchmarks.
* The paper is well-structured and easy to follow.

**Weaknesses:**

* Limited technical contributions. This work seems to be a direct application of Gumbel-Sigmoid, which is a general engineering trick (https://github.com/AngelosNal/PyTorch-Gumbel-Sigmoid) to approximate discrete sampling in a differentiable manner. It needs to be clarified what the main technical insights this work aims to bring to the 3DGS compression community. To me, replacing the threshold setting with the Gumbel-Sigmoid is too intuitive to hit the standard of a top-tier conference.

* It is difficult to know whether the learned binary mask achieves the “optimal Gaussian point size”. This work demonstrates that it learns the “optimal” solutions by plotting several line charts without any theoretical analysis. In some cases, LP-3DGS did not find the best balance, e.g., Fig. 4 Mini-Splatting on Kitchen. Besides, LP-3DGS seems to lose the flexibility of finding better solutions compared to its baseline, e.g., RadSplat can set a smaller ratio to compress further while maintaining reasonable visual quality.


* Lack of discussions with some recent related works. Several more recent approaches in the 3DGS compression community need discussion, and it would be better to add some comparisons with them to understand the advantages of LP-3DGS better.
  * Girish, Sharath, Kamal Gupta, and Abhinav Shrivastava. "Eagles: Efficient accelerated 3d gaussians with lightweight encodings." arXiv:2312.04564.
  * Lu, Tao, et al. "Scaffold-gs: Structured 3d gaussians for view-adaptive rendering." CVPR2024.
  * Chen, Yihang, et al. "HAC: Hash-grid Assisted Context for 3D Gaussian Splatting Compression." arXiv:2403.14530.

**Questions:**

Please kindly refer to the [Weaknesses].

**Limitations:**

Please kindly refer to the [Weaknesses].

---

> ### Author Rebuttal · Authors · 2024-08-07
>
> ### Response to weakness 1:
>
> Please refer to the Common Question about the term "contribution" for more details.
>
> ### Response to weakness 2:
>
> Please refer to the Common Question about the term "optimal" pruning for more details.
>
> ### Response to weakness 3:
>
> We have comprehensively compared our method with two of the most recent and state-of-the-art (SoTA) 3DGS pruning works, namely RadSplat and Mini-Splatting. The experimental results are reported in Figure 4, Tables 1 and 2, and additional results in the appendix. We appreciate the reviewer’s comment and will include comparisons with the suggested works.
>
> EAGLES primarily modifies the hyperparameters of the densification stage of 3DGS to control the number of Gaussians, which differs from score-based pruning methods such as RadSplat, Mini-Splatting, and LightGaussian. Our method uses a learnable mask on the scores of Gaussian points, making it incompatible with EAGLES.
>
> Scaffold-GS and HAC focus on encoding the attributes of 3DGS to compress the model, which is different from score-based pruning methods. These works use encoding methods rather than pruning, making them incompatible with our approach.
>
> We will cite these works as references and discuss them in the manuscript.

---

> > ### Author Response · Authors · 2024-08-13
> >
> > Dear reviewer, we hope our rebuttal has effectively addressed your comments and concerns. As the deadline for the author-reviewer discussion is tomorrow, we kindly request your prompt feedback on our response and ask you to reconsider the review score.
> > Thank you very much for your time and consideration.

---

> > ### Author Response · Authors · 2024-08-13
> > **last day of rebuttal response**
> >
> > Dear reviewer CWFJ,
> >
> > today is the last day of reviewer-author discussion period. We hope you have checked our rebuttal and found it could address your previous concerns. We would be happy to provide clarification of our work if you have any further comment.
> >
> > thanks for your review.

---

> ### Author Response · Authors · 2024-08-11
>
> Dear reviewer,
> We hope our rebuttal can address your comments and concern. As the deadline for the author-reviewer discussion is approaching, would you please share your thoughts on our rebuttal and reconsider the review score? thanks a lot

---

### Official Review · Reviewer_fvLt · 2024-07-13

**Soundness:** 3
**Presentation:** 2
**Contribution:** 2
**Rating:** 6
**Confidence:** 3

**Summary:**

This paper proposes a method to optimally prune Gaussians that do not participate in the rendering or optimizing process of the 3D Gaussian Splatting 3D reconstruction algorithm. The key idea in this paper is to use a Gumbel-sigmoid function instead of the standard Sigmoid function to optimize a binary mask based on transmittance of each splatted Gaussian. Experimental results show that by doing so the number of Gaussians can be reduced without losing reconstruction quality, keeping a near optimal balance between quality and computational speed. As a result, higher frame rate can be obtained for rendering.

**Strengths:**

-	The idea to use the Gumbel-function is interesting because it allows to naturally binarize the input values while keeping differentiability.
-	Results show that higher frame rate can be achieved without losing accuracy or training time. The ablation study is also good because it shows the advantage of optimizing the mask values compared to hard thresholding.

**Weaknesses:**

-	Except for replacing the sigmoid function with the Gumbel-Sigmoid function it seems that there is no other novel technical point in the paper. From the results it seems that a threshold value of 0.6 would be better than the naïve 0.5. I am wondering how the ablation study changes if using different threshold.
-	In the experiments, no comparisons to other methods that focus on optimizing number compressing the model. For example, comparison with LightGaussian would be interesting. It seems from the Figure 4 that the proposed method performs slightly worse than mini splatting.
-	What is the meaning of the “pruning ratio” for the proposed method in Figure 4? It seems to be different for the different datasets. However, as far as I understood, the proposed method does not use a pruning ratio: l. 194 => pruning […] points with mask values of 0. This is confusing.
-	There is no significant gain in accuracy, training time or memory consumption. The only significant gain is in rendering speed. This is interesting but because original 3DGS is already quite fast, in practice I am not sure the impact is high. For example if using a simple threshold of 0.5 in Compact3D how does frame rate compares?
-	Writing can be brushed up.
o	L.176-178 is not correct.
o	In figure 4, it seems legends are wrong in the lines for Mini-splatting => blue is described as RadSplat.
o	Tables are not easy to read. Add some special fonts like bold or underline to emphasize best and second-best results.

**Questions:**

-	Clarify Figure 4 and explain what the meaning of the pruning ratio for the proposed method is.
-	Add more comparative experiments with other compact 3DGS methods.

**Limitations:**

-	Limitations are properly discussed in the conclusion section.

---

> ### Author Rebuttal · Authors · 2024-08-07
>
> ### Response to weakness 1\&3:
>
> Please refer to the Common Question about the clarification and summary of our technical contributions.
>
> We would like to highlight that the main contribution of our work is not merely adopting the Gumbel-Sigmoid to prune 3DGS models but rather developing a learning framework that can automatically learn a pruning ratio embedding during the training process of a 3DGS model. This is achieved through our proposed Gumbel-Sigmoid mask pruning technique (Equation 8) and ensemble loss function (Equation 10), with only one round of training.
>
> Regarding the analysis shown in Figure 4, RadSplat/Mini-Splatting calculates their defined importance score for each Gaussian. If the target is to prune 60% of the entire model, the Gaussians are ranked based on this predefined importance score to achieve the 60% pruning. The actual pruning threshold of the importance score value can then be calculated.
>
> To clarify, Figures 1, 4, and 6 display the pruning ratio of the entire model versus the rendering quality, rather than the actual threshold of the importance score. This is why the x-axis ranges from 0 to 1, where each point corresponds to the rendering quality with a pruning ratio ranging from 0.1 to 0.9 in our experiments. Each point represents the result of one round of training with a preset pruning ratio, sweeping from 10% to 90%.
>
> For different pruning methods, such as RadSplat and Mini-Splatting, the results indicate that there always exists a balanced pruning ratio region that achieves a large pruning ratio with almost no or negligible rendering quality drop. In contrast, our method can learn such a pruning ratio with only one round of training.
>
> As observed by the reviewer, our method can automatically learn balanced pruning ratios with respect to different scenes. In other words, our proposed method does not require a pre-defined pruning ratio or threshold like prior pruning works, but instead learns one. We believe this is the greatest contribution of our work, and our method will benefit the community.
>
> ### Response to weakness 2:
>
> We kindly disagree with the reviewer and believe that we have already provided a comprehensive comparison with two of the most recent and best-performing state-of-the-art (SoTA) works, namely RadSplat and Mini-Splatting. The experimental results are reported in Figure 4, Tables 1 and 2, and additional results in the appendix.
>
> In the paper "RadSplat: Radiance Field-Informed Gaussian Splatting for Robust Real-Time Rendering with 900+ FPS", arXiv:2403.13806, the authors compared their method with LightGaussian, demonstrating better rendering image quality and a smaller model size. Consequently, we adopted RadSplat as our baseline.
>
> We appreciate the reviewer’s comments and, as suggested, we have also included a comparison with LightGaussian as follows:
>
> |                            | **Synthetic NeRF**   |            |            | **MipNeRF360 and Tank & Temple** |            |            |
> |----------------------------|-----------------------|------------|------------|----------------------------------|------------|------------|
> |                            | PSNR                  | SSIM       | LPIPS      | PSNR                             | SSIM       | LPIPS      |
> | LightGaussian              | 32.725                | 0.965      | 0.0370     | 27.206                           | 0.761      | 0.217      |
> | Ours (RadSplat Score)      | 33.709                | 0.9694     | 0.03139    | 27.812                           | 0.863      | 0.189      |
> | Ours (Mini-Splatting Score) | 33.496                | 0.966      | 0.0335     | 27.467                           | 0.856      | 0.201      |
>
> Regarding the comparison with Mini-Splatting, since our method is a learn-to-prune approach, it exhibits slight variations across different scenes. The average performance is reported in Table 1. It appears that the method performs better with the importance score defined in RadSplat compared to the one defined in Mini-Splatting, given our hyperparameter settings. We will investigate this further in our future work.
>
> ### Response to weakness 4:
>
> We kindly disagree with the reviewer on this comment. Please refer to our response to the Common Question about technical contributions for a detailed explanation. Our primary goal is not to simply learn a smaller model with improved performance as in previous works, but to develop a learning-to-prune framework that automatically determines a balanced pruning ratio. This approach achieves an excellent compression ratio without compromising rendering quality, addressing a challenge that has not been previously tackled in the 3DGS community.
>
> The overall training cost for our learning-to-prune methodology is significantly lower compared to prior manual pruning works, such as RadSplat, Mini-Splatting, and LightGaussian. Even when compared to Compact3D, our method, which uses a differentiable mask rather than a straight-through estimator (STE) binary mask, demonstrates better performance. Tables 4 and 5 present the ablation study that highlights this improvement.
>
> ### Response to weakness 5:
>
> We appreciate the comment. We will correct the image.

---

> ### Author Response · Authors · 2024-08-11
>
> Dear reviewer,
> We hope our rebuttal can address your comments and concern. As the deadline for the author-reviewer discussion is approaching, would you please share your thoughts on our rebuttal and reconsider the review score? thanks a lot

---

> > ### Comment · Reviewer_fvLt · 2024-08-12
> >
> > I have read the authors' response and the other reviews.
> >
> > Thanks for the detailed answers.
> > I had similar question as reviewer o7SR about W2. Figure 1-4 kind of gave the impression that the pruning ratio can be decided. It is now clarified that the parameter is the importance score, which results in some pruning ratio. And the proposed method allows to find the score to achieve desired pruning ratio.
> > Other comments have also been addressed the rebuttal. I raise my rating to 6.

---

### Official Review · Reviewer_o7SR · 2024-07-14

**Soundness:** 3
**Presentation:** 3
**Contribution:** 3
**Rating:** 6
**Confidence:** 4

**Summary:**

Whereas other works set a fixed pruning ratio or threshold, this paper introduces a method to learn how to prune Gaussians from a scene while retaining high image fidelity. Different scenes have a drop in fidelity at different pruning percentages and the goal of this work is to remove the need to train the model multiple times while searching for a pruning percentage near, but before, that drop. Specifically this work uses a Gumbel-Sigmoid — which is a 2-class Gumbel softmax followed by a Sigmoid function —  to enable a differentiable mask that can learn to prune Gaussians during training. This mask is then applied to set the opacity of each Gaussian. Gaussians where the mask goes to 0 are pruned. Evaluations are performed on 2 real world datasets — MipNerf 360 and Tanks & Temples — and one synthetic dataset — NeRF-Sythetic.

**Strengths:**

S1: The writing in this work is quite clear. The methodology is straightforward to understand and the explanations appear to be detailed enough to be able to replicate experiments

S2: This method seems useful to practitioners who are low on resources and want to quickly fit and compress a 3D-GS model.

**Weaknesses:**

**Main Weaknesses:**

W1: This paper uses the terms “optimal” and “best” to describe the pruning ratio found by learning. Specifically a soft definition of the optimal pruning ratio is given in line 41: “minimize the number of Gaussian points while keeping the rendering quality.” But this method does not “keep” the rendering quality — image quality metrics decrease after pruning. Generally the terms “optimal” and “best” are used to describe maximal or minimal points in some meaningful measure. In the domain of pruning 3D-GS models, a more appropriate definition of optimal pruning would be the highest possible compression that can be applied to the model while retaining rendering quality to some percent of the original rendering quality.

W2: It would be helpful to also provide analysis on pruning threshold vs. image quality metrics. It’s possible that there exists a pruning threshold for RadSplat and/or Mini-Splatting that prunes a large percent of Gaussians while retaining image quality.

W3: Table 2: Minutes isn’t a great metric here. FLOPs or MACs over training would be a lot better. Other works that want to compare to this method will need to use exactly this manuscript’s architecture to replicate its results.



**Other minor considerations:**

MC1: There are still several hyper parameters that haven’t been ablated. This is mostly fine, but it would be good to know how big of an impact these parameters have:
- Gumbel $\tau$
- $\lambda_m$
- Applying the pruning earlier/later than 20000 epochs

MC2: It would be interesting to ablate equation 8 by removing $S_i$. Can you simply just learn $m_i$ parameter without computing an importance score?


**Cosmetic issues:**

C1: line 46, line 50, line 62: could -> can

C2: $T$ in eq. 7 should be a $\tau$

C3: Tables 1 and 2, move the metric to the first column for clarity (like tables 4 and 5)

**Questions:**

My biggest concern for this manuscript is that the use of terms “optimal” and “best” is misleading. This method can find a good pruning ratio in one-shot, but it’s not clear that this ratio is optimal. I’m rating this manuscript a borderline reject for now because I strongly think this wording needs to change. I will raise my score if the authors can provide a detailed description of how they will update this wording in their rebuttal.

I would also like to see W2 and W3 addressed, but these are likely computationally expensive and aren’t as important to me as W1.

**Limitations:**

Once the wording in this manuscript is changed, I think the main limitations will be addressed.

---

> ### Author Rebuttal · Authors · 2024-08-07
>
> ### Response to W1:
>
> Thank you for the comment. Please refer to the Common Question about the term ``optimal" pruning.
>
> ### Response to W2:
>
> For Radsplat/Mini-Splatting, they calculate their defined importance score for each Gaussian. If the target is to prune 60% of the entire model, the Gaussians are ranked based on this predefined importance score to prune 60% off. Correspondingly, the actual pruning threshold of the importance score value can be calculated.
>
> To clarify, Figures 1, 4, and 6 show the pruning ratio of the entire model versus the rendering quality, rather than the actual threshold of the importance score. These figures already report what the reviewer asked for. Each point in these figures represents the result of one round of training with a preset pruning ratio, sweeping from 10% to 90%. The results indicate there always exists a balanced pruning ratio region for both Radsplat/Mini-Splatting that can achieve a large pruning ratio with almost no or negligible rendering quality drop. We have never denied this. However, our method can learn such a pruning ratio with only one round of training.
>
> ### Response to W3:
>
> Following the same training time metric used in most prior 3DGS model pruning works, such as RadSplat, Mini-Splatting, and EAGLES, we also use training time as an important metric for a direct comparison. All experiments we reported are conducted with the same GPU hardware setup, ensuring consistency. Our method learns a smaller 3DGS model, where the saved MACs and FLOPS depend on the learned pruning ratio.
>
> The training time we report is for only one round of training. This is to show that our proposed method incurs almost no training time overhead or even results in shorter training time due to the smaller pruned model in the later stages of the training pipeline. It is important to note that prior manual pruning works require multiple rounds of training to find the balanced pruning ratio setting, making it challenging to simply report the MACs and FLOPS.
>
> ### Response to MC1:
>
> Thank you for the comments. We conducted further ablation studies on those hyperparameters.
>
> We discovered that the parameter $\tau$ has a very minor effect on the learned pruning ratio.
>
> We also conducted an ablation study on the parameter $\lambda_{m}$ using the Room scene. The experimental results are as follows:
>
> Ablation Study on Parameter $\lambda_{m}$ (Room Scene)
> | $\lambda_{m}$ | PSNR  | SSIM  | LPIPS | Pruning Ratio |
> |---------------|-------|-------|-------|---------------|
> | 3e-4          | 31.488| 0.9241| 0.2035| 73.15%        |
> | 5e-4          | 31.490| 0.9243| 0.2032| 73.78%        |
> | 7e-4          | 31.474| 0.9238| 0.2040| 75.91%        |
>
> These results indicate that varying $\lambda_{m}$ has a negligible impact on the PSNR, SSIM, and LPIPS metrics in the Room scene.
>
> Regarding the timing of pruning the model, we found that the best time to perform pruning is at the 20,000th epoch. This timing allows the model extra training epochs to fine-tune based on the pruned model. Pruning the model too early is not advisable, as the 3DGS densification stage stops at the 15,000th epoch, and the model requires additional time to learn the scene. We have already reported the complete set of hyperparameter settings and open-sourced our code on GitHub for researchers to reproduce our results. We will include this ablation study in the final manuscript if space permits.
>
> ### Response to MC2:
>
> Given the rapid development of the 3DGS community, new pruning methods are emerging quickly. We hope our method is compatible with most pruning methods, especially those defining new importance scores for pruning (please refer to the Common Question about Contribution above).
>
> ### Cosmetic issues:
>
> We appreciate your comments and will address all cosmetic issues in the updated manuscript accordingly.

---

> > ### Comment · Reviewer_o7SR · 2024-08-08
> >
> > W1:
> > Thanks for the detailed reply. I think it’s fine if you include a different adjective to sell this method, just not “optimal”. For instance I think it’s totally fine to claim this is a “learned *effective* pruning ratio”.
> >
> > W2:
> > I think there may be some confusion here. I’m referring to the actual score computed by these methods, $s \in \mathbb{R}$. A score is computed per Gaussian, then these scores are ordered to prescribe, for a given pruning ratio $x$%, which of them should be pruned. It’s not possible, given only the Gaussian’s ordering (i.e., the pruning ratio) to obtain the original scores.
> >
> > This criticism was meant to call out that all plots are relative to this percentage, but not the actual scores. It’s possible that for one scene a score $s$ may be at the $40$% pruning ratio whereas that *same* score may be at the $60$% pruning ratio in a different scene. It’s entirely possible that a specific score may provide a better threshold across scenes because it likely will not corespondent to a specific pruning ratio, but will be a measure of the confidence in a particular Gaussian.
> >
> > W3:
> > I agree other methods report minutes. I am not overly concerned by this and don’t require the authors to run this.
> > Note: by FLOPs above I mean “floating point operations”, not FLOPS “floating point operations per second”.
> >
> > MC2:
> > This comment was that it would be interesting to ablate how well this method performs *without* the use of an auxiliary pruning score. It’s possible that $m_i$ may prove to be a good pruning score on its own. I suspect that $S_i$ may be only providing a good “initialization”.
> >
> > Overall:
> > I still find this method interesting and my primary concern, the wording of the manuscript, has been addressed.

---

> > > ### Comment · Reviewer_o7SR · 2024-08-11
> > >
> > > Apologies to the authors, I meant to raise my score to a 5 earlier based on the rebuttal. I'm happy to raise my score to a 6 if my concern regarding W2 can be addressed.

---

> > > > ### Author Response · Authors · 2024-08-12
> > > >
> > > > We appreciate all the reviewer’s valuable comments and insights, which have genuinely helped improve our work. We would be grateful if the reviewer could consider raising the final score.

---

> > > > > ### Comment · Reviewer_o7SR · 2024-08-12
> > > > >
> > > > > Interesting, yes this addresses my concern regarding W2. I'll update my score to a 6.

---

> ### Author Response · Authors · 2024-08-12
>
> # W1
> We appreciate your comment. we will change to learned effective pruning ratio.
>
> # W2
> We thank you for your further clarification. Please see our detailed response below:
>
> ***"It’s possible that for one scene a score may be at the 40% pruning ratio whereas that same score may be at the 60% pruning ratio in a different scene."***
>
> We totally agree with the reviewer. To justify such, we conducted extra experiments to get the pruning ratio while fixing the pruning threshold of RadSplat defined importance score. The results are listed in below table. It can be clearly seen that the same threshold score for different scenes lead to different pruning ratios for different scenes. This actually further justifies the importance of our work, as it implies that different scenes indeed need the "learned effective pruning ratio'' to optimize tradeoff between compression ratio and rendering quality.
> Due to the fact that it is a most common setting in prior works to measure the rendering image quality with target pruning ratios, for fair comparison, we design our framework to learn an effective pruning ratio for each scene, without multiple rounds of manual tuning of pruning ratios as used in prior works
>
> | Threhold | Bicycle | Bonsai | Counter | Kitchen |  Room  |  Stump | Garden | Flowers | Treehill |
> |:-----:|:-------:|:------:|:-------:|:-------:|:------:|:------:|:------:|:-------:|:--------:|
> | 0.025 |  50.88% | 43.79% |  47.86% |  38.46% | 52.73% | 51.07% | 48.05% |  50.65% |  41.26%  |
>
> ***"It’s entirely possible that a specific score may provide a better threshold across scenes because it likely will not corespondent to a specific pruning ratio, but will be a measure of the confidence in a particular Gaussian."***
>
> This is a very interesting point. We conducted such experiments on different scenes using different threshold values, based on RadSplat pruning importance score and the results are shown in figure below. It can be seen that, in the kitchen scene, the PSNR starts to drop significantly when the threshold is larger than 0.2. However, in the room scene, such cutoff point is around 0.15. It shows different scenes have different sensitivity against pruning threshold.
>
> [Kitchen](https://drive.google.com/uc?export=view&id=1dTP7mA_MIw_ROhxstItbThJyI6m-Sr3L)
> [Room](https://drive.google.com/uc?export=view&id=1qU1gwcNFo6hwcDOC7qch71YZcTcET0rU)
>
> More interestingly, since our proposed method learns the effective pruning ratio for different scenes during 3DGS model construction, we discovered that it naturally captures such unique cutoff threshold score for different scenes.
>
> # MC2
> We appreciate you for sharing this valuable insight.
> In equation 8, if we only use $m_i$, then we could regard opacity ($o$) as the importance score. It is also true that the opacity itself could be one type of importance score. Our objective is to propose a method that is general, learnable, and orthogonal to other prior or potentially future works exploring different definitions of effective 3DGS pruning importance scores.

---

### Author Rebuttal · Authors · 2024-08-07

## Common question about the term "optimal" pruning (Reviewer o7SR and Reviewer 1UP2 )

We agree with Reviewer o7SR that the term "optimal" may be misleading. We will change it to "learned pruning ratio" and provide a more detailed definition in the updated manuscript. In the domain of 3DGS model pruning, the primary objective is to compress the model size as much as possible using a predefined importance factor, without significantly affecting rendering image quality, which involves a trade-off process.

As demonstrated in Figure 4, which shows the rendering quality versus pruning ratio, a small pruning ratio typically does not hamper rendering quality due to model redundancy. However, a very large pruning ratio will significantly degrade the rendering quality. There exists a small region that can achieve a relatively large pruning ratio with almost no or slightly degraded rendering quality, which is the optimal or more balanced pruning ratio range we aim to target.

In all existing works on 3DGS model pruning, the optimal or final reported balanced pruning ratio is usually determined empirically by manually sweeping the pruning ratio from low to high (e.g., 0.1 to 0.9). Each predefined pruning ratio sweep point requires one round of model training, necessitating multiple rounds of pruning-aware training to optimize or balance the compression (i.e., pruning) ratio and rendering quality.

In contrast, our proposed methodology can automatically learn a pruning ratio embedding during the training process of the 3DGS model through our proposed Gumbel-sigmoid mask pruning technique and ensemble loss function (Equation 10), with only one round of training. The experiments in Figure 4 and other reported tables clearly show that our method consistently learns an equivalent or even better pruning ratio compared to manual tuning in existing works.

## Common question about technical contributions (Reviewer fvLt and Reviewer CWFJ)

As discussed in the previous common question, identifying the optimal or balanced pruning ratio in prior works is both compute- and time-consuming. We would like to highlight that the main contribution of our work is not merely the adoption of the Gumbel-sigmoid to prune 3DGS models, but rather the development of a learning framework that can automatically learn a pruning ratio embedding during the training process of a 3DGS model. This is achieved through our proposed Gumbel-sigmoid mask pruning technique (Equation 8) and ensemble loss function (Equation 10), all with only one round of training.

Specifically, to distinguish our approach from prior pruning works in 3DGS, the advantages of the proposed method are:

* Highly Efficient: Identifying balanced pruning ratio candidates from 0.1 to 0.9 with a 0.1 sweeping resolution typically requires 9 rounds of training. In contrast, our proposed method requires only one training round, reducing the overall training cost by up to 9x.

* General: Our proposed Gumbel-sigmoid mask pruning technique can be applied to a general importance score (e.g., opacity, or any other importance score indicating the importance of Gaussian) as defined in Equation 8. This means it is orthogonal and compatible with the pruning criteria defined in prior and potentially future 3DGS model pruning works.

* Effective: Our experimental results strongly demonstrate that our proposed learning-to-prune method successfully learns a pruning ratio that is always within the balanced pruning range, maintaining a good trade-off between compression ratio and rendering quality compared to existing works.

---

### Decision · Program_Chairs · 2024-09-25

**Decision:**

Accept (poster)

**Comment:**

This paper received mixed scores (two Weak Accepts, Borderline Reject, and Borderline Accept) from 4 reviewers. Three reviewers (o7SR, fvLt, 1UP2) are mainly satisfied with the responses from the authors during the rebuttal phrase. Reviewer CWFJ initially suggests Borderline Reject and has concerns in technical contribution, optimal Gaussian point size, and discussion with existing works. During the rebuttal phase, the reviewer did not response. These concerns are shared by other reviewers and are believed that they are clarified. The AC agrees with the majority of reviewers that the work can be accepted. The authors are encouraged to include these important clarifications into the final paper when preparing the camera ready version.